# Chloroplasts— Beyond Energy Capture and Carbon Fixation: Tuning of Photosynthesis in Response to Chilling Stress

**DOI:** 10.3390/ijms20205046

**Published:** 2019-10-11

**Authors:** Ping Gan, Fang Liu, Rongbai Li, Shaokui Wang, Jijing Luo

**Affiliations:** 1College of Life Science and technology (State Key Laboratory for Conservation and Utilization of Subtropical Agro-bioresources), Guangxi University, Nanning 530004, China; 2Agriculture College, Guangxi University, Nanning 530004, China; 3Agriculture College, South China Agricultural University, Guangzhou 510642, China

**Keywords:** chloroplast, chilling stress, regulatory response, photosynthesis, redox homeostasis, hormones

## Abstract

As organelles for photosynthesis in green plants, chloroplasts play a vital role in solar energy capture and carbon fixation. The maintenance of normal chloroplast physiological functions is essential for plant growth and development. Low temperature is an adverse environmental stress that affects crop productivity. Low temperature severely affects the growth and development of plants, especially photosynthesis. To date, many studies have reported that chloroplasts are not only just organelles of photosynthesis. Chloroplasts can also perceive chilling stress signals via membranes and photoreceptors, and they maintain their homeostasis and promote photosynthesis by regulating the state of lipid membranes, the abundance of photosynthesis-related proteins, the activity of enzymes, the redox state, and the balance of hormones and by releasing retrograde signals, thus improving plant resistance to low temperatures. This review focused on the potential functions of chloroplasts in fine tuning photosynthesis processes under low-temperature stress by perceiving stress signals, modulating the expression of photosynthesis-related genes, and scavenging excess reactive oxygen species (ROS) in chloroplasts to survive the adverse environment.

## 1. Introduction

Higher plants are of great importance because of their roles in solar energy capture and carbon fixation, providing energy and biomass for the whole biosphere. However, higher plants encounter adverse abiotic stresses such as chilling/freezing, drought, salinity, and heat stresses as well as biotic stresses caused by bacteria, viruses, fungi, parasites, and beneficial and harmful insects and thus have evolved a number of strategies to cope with these stresses. However, global warming and climate change have caused frequent occurrences of extreme weather conditions that can impair plant growth and development.

Low-temperature conditions, including chilling and freezing, are environmental stresses that can hinder plant growth and development. This review focuses on the response mechanisms of chloroplasts to chilling stress because many crop-plants experience this during the growing season. Chilling stress occurs in the temperatures range 0–15 °C [1]. Many food and economically important crop species that originate from tropical and subtropical regions are thermophilic and sensitive to low temperatures. For example, for major food crop species such as rice and maize and one of the world’s major economically important crop species, tobacco, low temperature is an important constraint on the growth, yield and distribution of those species. At low temperatures, rice seedlings display slow development, reduced numbers of tillers, yellowish leaves and even wilted leaves [2]. Chilling stress also has adverse effects on panicle growth and fertility of rice plants at the reproductive stage, which leads to tremendous yield losses [3]. Chilling temperatures limit rice yields in mountainous, tropical regions and in temperate rice-growing zones. In approximately seven million hectares in South and South-East Asia, including high altitude regions in Bhutan, Burma, India, Indonesia, Nepal, the Philippines, and Vietnam, modern rice varieties cannot be planted due to the cool climate inherent to these regions [4]. However, after a long period of natural selection, a series of physiological, biochemical and molecular mechanisms have evolved in plants to cope with changing environmental conditions, including chilling stress. Studies have shown that plants undergo a series of physiological and cellular regulatory activities at low temperatures, including alterations to the membrane structure, photosynthesis, calcium signaling and metabolism, to help them adapt to cold environments [1].

Chloroplasts constitute one of the most important organelles of green plants and are the only location for the biochemical process of photosynthesis. Chloroplasts capture light energy to assimilate carbon dioxide (CO_2_) and water, synthesize energy-storing organic matter, and produce oxygen (O_2_) (Figure 1). Chloroplasts are biological factories with the lowest cost of biomass production worldwide. In addition, many other metabolic processes, including the synthesis of lipids, pigments, plant hormones, and other metabolites, occur in chloroplasts. Therefore, normal physiological functions of chloroplasts are essential for plant growth and development.

Photosynthesis is a complex, multi-step process comprising many biological pathways. The photoelectron transport system and the Calvin cycle are two important steps in photosynthesis that involve the conversion of light energy into adenosine triphosphate (ATP) and nicotinamide adenine dinucleotide phosphate (NADPH) and the fixation of CO_2_ into carbohydrates. These two processes are controlled by many genes/gene products encoded by the chloroplastic and nuclear genomes [5,6]. The expression of genes related to photosynthesis in the chloroplast and nucleus is highly variable. The gene products work together in an extensively coordinated photosynthesis network composed of genes, regulatory components, signaling factors and metabolic processes and are affected by various environmental factors [7]. Abiotic stresses such as drought, salinity, flooding, high light, high temperature and cold can restrict the process of photosynthetic carbon metabolism in plants and can adversely affect plant growth and productivity. In agriculture, various abiotic stresses have become the main limiting factors for crop production. Abiotic stresses are ultimately defined as those that impact crop yields, i.e., the final economic output [8]. One of the non-negligible abiotic conditions is temperature stress, which is the main environmental factor that affects the duration of a plant’s growing season as well as its biogeographical distribution [9], limiting crop yields.

Chloroplasts are highly sensitive to chilling stress, which inhibits photosynthesis. Moreover, the photosynthetic carbon metabolism of plants exposed to non-optimized low temperature is severely affected. This stress can not only damage the ultrastructure of organelles, but also alter pigments and the concentration of metabolites in chloroplasts and stomatal regulation [10]. There is growing evidence that chloroplasts play a pivotal role in cellular responses to signals, generating a variety of signals to coordinate the fine tuning and appropriate responses to all particular situations [11]. In fact, the hormones abscisic acid (ABA), jasmonic acid (JA) and salicylic acid (SA), which are produced by chloroplasts, as well as reactive oxygen species (ROS) and redox signals, are critical components of the plant stress response. Therefore, the chloroplast response is a key regulatory point for the survival of plants under abiotic stress. Additionally, chloroplasts can perceive external stimuli and respond to external stresses promptly. Furthermore, retrograde stress signals are generated by chloroplasts and regulate the expression of stress-related nuclear genes to coordinately tune cellular responsive processes in plants to adapt to adverse environmental conditions.

Here, the authors briefly review how chloroplasts perceive chilling stress, highlight the responses of chloroplasts to chilling stress via complex signaling pathways and discuss the potential functions of chloroplasts in addition to energy capture and carbon fixation. Those are the cellular response processes that occur in chloroplasts, including the modulation of photosynthesis under chilling stress, which act as a fine-tuning regulatory mechanism.

## 2. Perception of Chloroplasts to Chilling Stress Signals

Chloroplasts can sense low temperatures and are the earliest organelles in plants to be severely affected by low temperatures [12]. The thylakoid membranes of chloroplasts contain photosystem complexes, which are also known as photosynthetic membranes, that are the locations of the light reactions of photosynthesis. Low temperature alters the composition and structure of the photosynthetic membrane, rigidifying the photosynthetic membrane and slowing the enzymatic reactions [13]. The degradation of photosynthetic membrane-coupled proteins and the inhibition of protein synthesis damage the integrity of photosynthetic membranes, thereby impairing the light reactions with photosynthetic membranes. It has been reported that both the membranes and photoreceptors can sense abnormal temperature changes. Plant cells perceive chilling stress usually by the reduction in membrane fluidity [14,15], which may be detected by various integral membrane proteins, including channel transporters and other various transporters, and membrane-anchored receptor kinases [16]. Recent studies have demonstrated that the photoreceptor phyB and phototropin are thermosensitive molecules that sense environmental temperature fluctuations [17,18,19]. As a component of the cold-avoidance response regulatory mechanism, phototropin 2, a blue light receptor, has been found to induce the localization of chloroplasts in response to cold under blue light, thereby optimizing photosynthesis during chilling stress [20]. Therefore, the physiological and biochemical responses of chloroplasts in response to chilling stress may be sensed by lipid membrane systems and photoreceptors.

## 3. The Regulatory Response to Chilling Stress Signals in Chloroplasts

In the chilling response, membrane proteins and membrane-anchored proteins of the chloroplasts perceive the stress signals, which are transmitted across the membrane into the chloroplast. Low temperature directly affects the structure and activity of the plant photosynthetic apparatus. First, the lipid membrane state, the abundance of photosynthesis-related proteins and the activity of numerous enzymes, are regulated in response to chilling stress. The photosynthetic efficiency is then downregulated, and excess ROS are produced [21,22,23]. As specific sensors and regulators of intracellular and extracellular stimuli, chloroplasts involve the integration of a large number of intracellular signals and metabolic pathways to sustain homeostasis at both the cellular and organismal level under low-temperature conditions.

### 3.1. Regulation of the Lipid Membrane State of Chloroplasts

The fluidity of bio-membranes facilitates the transport and trans-membrane exchange of materials and contributes to energy flow and information transfer. When plants are exposed to chilling stress, the change in membrane fluidity, including that of chloroplastic membranes, is the earliest response in plant cells. However, the phase change of the membranes under chilling stress largely depends on the lipid composition and is closely related to the degree of unsaturated fatty acyl chains.

Monogalactosyldiacylglycerol (MGDG) and digalactosyldiacylglycerol (DGDG) are the two nonionic lipid constituents of the thylakoid membrane of higher plants and indispensable for photosynthesis. MGDG, a non-bilayer-forming lipid, accounts for approximately 50 mol% and DGDG, a bilayer-forming lipid—25 mol% of the thylakoid membrane lipids [24]. In thylakoid membranes, the ratio of non-bilayer lipids to bilayer lipids is generally constant, which is very important for maintaining the normal fluidity of thylakoid membranes. The increased level of DGDG in dark-chilled cucumber cotyledons and thus diminishing of the MGDG/DGDG ratio was evidently responsible for a decrease in membrane fluidity [25].

The genetic manipulation of fatty acid (FA) unsaturation has been shown to alter the sensitivity of transgenic plants to low temperature [26]. The level of unsaturated fatty acids (USFAs) is regulated by the activity of fatty acid desaturase (FAD) [27]. At low temperatures, the polyunsaturated fatty acid (PUFA) content in cells increases to maintain proper membrane fluidity and thus growth under chilling stress [28]. For example, in a study of the role of the ω-3 FA desaturase gene *OsFAD8* in the cold acclimation process of rice, it was found that the USFAs in the thylakoid membranes of *OsFAD8* knock-out mutants were significantly decreased, leading to a significant reduction in membrane fluidity, which implies that FA unsaturation regulated by *OsFAD8* is crucial for higher plants to adapt to chilling stress [29]. Furthermore, when tobacco was exposed to 8 °C for several days, the relative levels of USFAs in the chloroplast envelopes and thylakoid membranes increased significantly, thereby altering the fluidity of the membrane to accommodate the high-level function of the photosynthetic apparatus [30,31,32]. As mentioned above, the presence of high levels of USFAs in the chloroplast membranes increase plant tolerance to low temperatures [26,33,34]. Thus, the lipid composition has a profound effect on the physical properties of a membrane. Once lipid homeostasis is disrupted, the corresponding membrane function will be impaired, and chloroplast morphology will be negatively affected.

### 3.2. Regulation of the Photosynthesis-Related Protein Abundance

Our recent report and many previous studies showed that the abundance of various proteins involved in photosynthesis, including chlorophyll a/b-binding protein (CAB), photosystem II (PSII) reaction centre P680 chlorophyll A apoprotein, the oxygen-evolving complex (OEC) proteins and oxygen-evolving enhancer (OEE) proteins, are regulated actively or passively under low-temperature conditions. Together with photosynthetic pigments, CAB form light-harvesting complexes (LHCs), which capture light energy and transfer it to the PSII reaction centre P680 rapidly, at which point the photochemical reactions are initiated. In addition, LHCs are involved in maintaining thylakoid membrane structure and regulating the distribution of excitation energy between PSII and photosystem I (PSI) [35,36]. The OEC, which can split water, releasing O_2_, is an important member of PSII [37]. The OEE proteins are members of the OEC and play an important role in the absorption of light energy by the OEC. The abundance of CAB, chlorophyll A apoprotein, OEC and OEE proteins in rice was shown to be upregulated after treatment at 14 °C for 48, 72 and 96 h [38]. The increased abundance of these proteins in rice favours the maintenance of the photolysis of water to form O_2_ under low-temperature conditions.

The photosynthetic electron transport process is also affected by chilling stress. The cytochrome b6f protein complex (Cytb6f), a supramolecular complex involved in the primary reaction process of photosynthesis that is present on the photosynthetic membrane, is one of the major membrane proteins. It connects the electron transfer process from PSII to PSI, oxidizes plastoquinone, generates trans-membrane proton gradients, and catalyses the synthesis of ATP. In addition, ferredoxin-NADP reductase (FNR) catalyses the synthesis of NADPH and plays a key role in photosynthetic electron transport. In rice, after 48 and 96 h of 14 °C treatment, the abundance of Cytb6f increased, while the abundance of FNR decreased [38]. These results indicated that the release of O_2_ due to the photolysis of water in rice leaves was promoted under low temperatures and that the primary reaction of the photosynthetic electron transport chain was activated. However, the activity of the light reaction-related enzymes was inhibited by low temperatures, implying that photosynthesis was attenuated in response to chilling stress.

In our previous study, a time series of chilling stress treatments was used to explore the dynamic changes in protein abundance in two genotypes with contrasting chilling stress tolerance at the proteomic level. Among all the differentially expressed proteins (DEPs) identified in the study, in line with the numerous studies mentioned above, the expression of the photosynthesis-related proteins, including the core components of the PSI and PSII reaction centres such as psbA (P0C434), psbC (P0C367), psbB (P0C364), Q2QWN3, A0A0N7KJ79, psaA (P0C355), CAB1R (P12330), and Q943K1, was significantly downregulated in two genotypes, DC90 and 9311, under chilling stress. Moreover, interestingly, a cluster analysis of the dynamic changes in DEPs among the time points of chilling stress and the subsequent recovery stage revealed an obvious difference in the dynamic changes in DEPs between the two genotypes. In the chilling-tolerant genotype DC90, the abundance of photosynthesis-related DEPs similarly tended to decrease under chilling stress and return to normal levels during the subsequent recovery stage. However, in the chilling-sensitive variety 9311, the abundance of downregulated DEPs did not recover during the recovery stage. Our results therefore indicated that the cold acclimation and de-acclimation of chloroplastic proteins might be regulated at the proteomic level in rice to adapt to chilling stress [35].

### 3.3. Regulation of Dark Reaction-Related Enzyme Activities

Under low-temperature conditions, the Calvin cycle-related enzyme activities in the dark reactions of photosynthesis are inhibited, resulting in a decrease in the rate of CO_2_ assimilation. According to previous reports, under light–chilling conditions, the reductive activity of two key enzymes involved in the Calvin cycle, fructose-1,6-diphosphatase (FBPase), which controls the synthesis of sucrose, and isoheptanone-1,7-diphosphatase (SBPase), which controls the influx and regeneration of carbon, was significantly reduced [39,40,41]. As two important phosphatases, FBPase and SBPase are involved, not only in the synthesis of carbohydrates, but also in the regeneration of ribulose-1,5-bisphosphate (RuBP). Thus, these two enzymes are important rate-limiting enzymes in photosynthesis [42]. The decreased reductive activity of these two enzymes can slow intracellular anabolism and indirectly affect the consumption of NADPH and ATP produced during the light reactions, resulting in an overaccumulation of ATP and NADPH at the end of the light reactions. The obstruction of the energy flow from the light to the dark reactions leads to excess light energy absorbed by photosynthesis. This excess energy can destroy the photosynthetic apparatus, thus leading to a decline in the photosynthesis rate of plants.

Under low-temperature conditions, the activity of ribulose-1,5-bisphosphate carboxylase/oxygenase (Rubisco) is also downregulated. Rubisco has dual carboxylase and oxygenase activities. Rubisco’s carboxylase activity is the rate-limiting step of photosynthesis. The carboxylation efficiency of Rubisco determines the photosynthesis rate of plants. In addition, Rubisco’s oxygenation reaction participates in the process of photorespiration, which consumes excess photons and thus is one of the photoprotection pathways of plants [43]. Low temperature reduces the Rubisco carboxylase activity and reduces the oxygenase activity [41]. Numerous studies have shown that chilling stress can destroy Rubisco proteins or can affect the redox regulation of many different Rubisco activase isoforms [40,41,44].

To adapt to low temperatures, plants synthesize additional photosynthetic enzymes involved in the photosynthetic carbon reduction cycle and sucrose synthesis, such as sucrose phosphate synthase, Rubisco, SBPase, stromal FBPase and cytosolic FBPase, to compensate for the reduced activities of other enzymes during chilling stress [45,46,47,48,49,50]. Therefore, photosynthesis strongly recovers after a certain period of time after plants are exposed to low temperatures.

### 3.4. Regulation of the Redox State of Chloroplasts

Chloroplasts are the main sites where ROS such as superoxide anion (O_2_^−^), hydrogen peroxide (H_2_O_2_), hydroxyl radical (
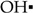
), and singlet oxygen (^1^O_2_) are generated due to the highly oxidizing metabolic activity of these compounds and increased electron flow rate [51]. The ROS in plants are in a dynamic equilibrium under optimal conditions and cannot severely damage plants. However, under stressful conditions, with the reduction in the ability of plants to utilize light energy, excessive harvested light energy is converted to ROS, thereby affecting the physiological and biochemical functions of plants [52]. However, plants have developed many strategies to cope with the overaccumulation of ROS and maintain normal photosynthetic efficiency. These processes involve complicated redox reaction chains and ROS-scavenging systems.

When the environmental temperature is substantially lower than the temperature for normal growth, the light and dark reactions of photosynthesis can be unbalanced, resulting in excess electrons within the transport chain. The photosynthetic electron transport chain transfers excess electrons to O_2_ as an alternative electron acceptor to release more O_2_^−^. For example, low temperature can cause an increase in O_2_^−^ in bermudagrass cells [53]. On the one hand, excess ROS are toxic to aerobic plants and can cause peroxidation of the chloroplast bio-membrane system and disrupt its function. Similarly, bio-membrane-anchored functional proteins can also be attacked by ROS. Moreover, continuous chilling stress disrupts the integrity of the bio-membrane and eventually leads to the loss of membrane system activity [54]. On the other hand, excess ROS promote the enhancement of the antioxidant system in plants, further reducing the damage caused by ROS to plants. However, aside from the toxicity of ROS indicated above, they are considered secondary messengers and play roles in regulating plant growth, development, and stress responses [55,56,57,58,59,60,61]. As stress signals, ROS can regulate gene expression in both the chloroplasts and nucleus, thereby regenerating damaged proteins, restoring lipid homeostasis and promoting plant adaptation to low temperature. ROS signaling may also regulate the expression of antioxidant-related genes by activating mitogen-activated protein kinase (MAPK) cascades [56].

The antioxidant system includes antioxidant enzymes and non-enzymatic components. The antioxidant enzymes include mainly superoxide dismutase (SOD), peroxidase (POD), polyphenol oxidase (PPO) and catalase (CAT), as well as ascorbate peroxidase (APX), dehydroascorbate reductase (DHAR), and glutathione reductase (GR), which play a key role in the ascorbic acid-glutathione (ASA-GSH) cycle. The non-enzymatic components include carotenoid (Cars), vitamin E (VE), ascorbic acid (ASA), glutathione (GSH), etc. [62]. SOD and the ASA-GSH cycle constitute the main ROS-scavenging mechanisms in chloroplasts. SOD catalyses the formation of H_2_O_2_ and H_2_O from two superoxide molecules and two protons [63]. Plants have several SOD genes, which are usually expressed in species-specific and stress-specific manners [64]. In the ASA-GSH cycle, using ascorbate as an electron donor, APX catalyses the reduction of H_2_O_2_ to water to form dehydroascorbic acid (DHA). The resulting DHA is recycled to ascorbate by DHAR, with GSH used as a reducing agent. Oxidized glutathione (GSSG) is subsequently regenerated by GR together with NADPH [63] (Figure 2). These two ROS-scavenging mechanisms are very important for maintaining the normal function of chloroplasts under chilling stress.

Previous studies have shown that the levels of ROS and the activities of ROS-scavenging enzymes increase in chloroplasts when plants are subjected to chilling stress [65,66,67]. Genetic manipulation of the components of antioxidant systems in chloroplasts can alter the tolerance of transgenic plants under low temperatures. The overexpression of Cu/Zn SOD and APX in chloroplasts of transgenic plants enhanced the tolerance of plants to high light and chilling stresses [23,68,69]. The knock down of the expression of the GR gene in the chloroplasts of antisense transgenic plants led to low-temperature sensitivity [70,71]. Therefore, maintaining redox homeostasis in chloroplasts is very important for plants to survive low-temperature stress.

It is generally believed that plants can alleviate stress-induced peroxidation damage by enhancing the activities of protective enzymes, thus increasing cold resistance. Several studies have shown that exogenous applications of methyl jasmonate, brassinolide, melatonin and acetylsalicylic acid can enhance the activity of the antioxidant system in chloroplasts of plants under low temperature. The enhancement of antioxidant systems in chloroplasts promotes the protection the chloroplast structure, promotes photosynthesis, and improves plant resistance to low-temperature stress [72,73,74,75].

### 3.5. Regulation of Retrograde Signals in Chloroplasts

Photosynthesis is an extremely complex process that is regulated by many genes/gene products encoded by both chloroplastic and nuclear genomes. The number of chloroplast proteins encoded in the nucleus is significantly greater than the number encoded by the chloroplast itself. There are 3000 to 4000 different proteins that localize to the chloroplast, most of which are encoded by nuclear genes; only a few are encoded by chloroplastic genes [76]. For example, in *Arabidopsis thaliana* chloroplasts, 2300 proteins are predicted to be encoded by the nuclear genome, while only 87 proteins are encoded by the chloroplast genome [77]. Therefore, a high degree of coordination between the nucleus and chloroplast is required to regulate target genes in response to chilling stress.

The chloroplast proteins encoded by the nucleus enter the chloroplast as anterograde signals. In turn, some chloroplast products/signaling molecules can be used as retrograde signals to regulate the expression of nuclear-coded genes. The disruption of chloroplast homeostasis induced by external stress can also be transduced to the nucleus by retrograde signals to coordinate cellular activity on the basis of the degree of chloroplast stress by tuning the expression of stress-induced genes and by modulating the alteration of cellular physiological, metabolic, and enzymatic activities, thereby regulating the supply of carbohydrates and other compounds [16].

Chilling stress under light can lead to the production of ^1^O_2_. ^1^O_2_ is capable of triggering a signaling pathway that involves two calcium-sensitive receptor proteins: EXECUTER1 (EX1) and EXECUTER2 (EX2). These two proteins are encoded by nuclear genes and are located on the thylakoid membrane [78]. Other reports have shown that signals triggered by ^1^O_2_ can also be activated by non-enzymatic oxidation products of β-carotene in an EX1- and EX2-independent pattern [79]. The retrograde ^1^O_2_ signal is transmitted to the nucleus and a variety of stress response pathways are induced to generate a series of secondary messengers, such as reactive electrophile species (RES). These are mainly derived from FA oxygenation products that contain α, β-unsaturated carbonyl groups or other reactive electrophilic atom groups that may modulate the expression of cell survival genes during severe stress (Figure 2) [80]. Therefore, all cellular responses can be regulated and integrated according to the degree of stress and ultimately produce stress response-related components, such as carbohydrates [51,81].

In recent years, various mechanisms of retrograde signals from the chloroplast to the nucleus have been well characterized [82]. Nuclear/plastid coordination and signal transduction mechanisms are highly important when plants are subjected to abiotic stresses, as such coordination mitigates damage to the photosynthesis systems of plants. Whether the chilling stress signals trigger other kinds of retrograde signals, such as methylerythritol cyclodiphosphate (MEcPP), 3-phosphoadenosine 5-phosphate (PAP), tetrapyrrole and other retrograde signals, remains unknown.

### 3.6. Regulation of Hormones in Chloroplasts

Plant hormones regulate various growth and development processes under normal growth conditions. Under stress conditions, hormones can trigger defence pathways to enhance specific stress tolerances. The major hormones that regulate plant responses to abiotic stresses are ABA, JA, SA, etc. [83].

ABA is a sesquiterpene hormone compound and is produced by the methylerythritol phosphate (MEP) pathway in plastids, which also produces Cars. ABA is an important hormone that mediates abiotic stress responses and plays an important role in regulating the physiological processes involved in stomatal opening in plants [84]. ABA signaling is particularly important during chilling stress. Low temperature induces an ABA accumulation in plants, after which the stomata close by a series of regulatory activities. After stomatal closure, with the reduction in the concentration of CO_2_ in cells, the rate of carbon assimilation and photosynthesis decrease [85]. Therefore, ABA can affect photosynthesis by regulating stomatal opening at low temperature.

JA is a lipid-derived hormone synthesized by the octadecanoid pathway. JA synthesis starts in the chloroplast but is completed in the peroxisome, after which a variety of metabolites with different functions are derived from the compound [86]. Recent studies have shown that JA plays a positive mediator role in cold tolerance and freezing tolerance [87]. Exogenous JA can increase the freezing tolerance of Arabidopsis, while *Arabidopsis* plants with mutations in JA synthesis or signal transduction defects display increased sensitivity to freezing stress [88].

SA is a phenolic hormone compound. One of its important synthesis pathways, the isochorismate synthase (ICS) pathway, is located in the chloroplast. In this pathway, isochorismate catalyses the conversion of chorismic acid to isochorismate and then catalyses the formation of SA via isochorismate pyruvate lyase (IPL) [89]. SA can be induced by a variety of environmental stresses, including chilling stress. After pretreatment with SA synthesis inhibitors in cucumber seedlings, endogenous SA accumulation decreased and was more susceptible to low-temperature stress [90]. The inhibition of SA synthesis under chilling stress resulted in the accumulation of H_2_O_2_ in cells. This indicates that SA signals are involved in both the regulation of cold-responsive gene expression and the elimination of H_2_O_2_ and play an important role in the response to chilling stress.

## 4. Conclusions and Perspectives

Environmental stresses are great influence factors to plant growth and development. When plants are subjected to abiotic stresses, the pathways of photosynthesis are severely inhibited. In crop species, the photosynthesis rate is reduced and the assimilated substances decrease significantly under chilling stress, which eventually leads to yield losses. The chloroplast thylakoid membrane and nucleus are the main targets of regulatory proteins and metabolites that are associated with the photosynthesis pathway. The rapid response of plant cell metabolism and the fine tuning of photosynthesis pathways are the key factors for plant survival under adverse environmental conditions.

It is well known that chloroplasts are a unique and indispensable organelle in green plant cells because they are essential for photosynthesis. Recent advances in chloroplast biology have shown that the functions of chloroplasts go far beyond photosynthesis and encompass all aspects of plant biology, including plant stress responses to abiotic stresses. In plant cells, chloroplast structure and the metabolic pathways that occur in those organelles are targets for fine tuning to maintain cellular redox homeostasis and for coordinating with other compartments of cells in response to chilling stress, thereby facilitating plant survival under adverse environmental conditions.

Chloroplast ultrastructure is also an important factor responsible for chloroplast function at low temperatures [91]. Normally, the thylakoids are arranged in parallel, the stroma lamellae are arranged regularly, the structure is clear in chloroplasts, and the chloroplast membrane is intact. However, under low-temperature stress, the morphological structure of chloroplasts obviously differs. For example, chloroplasts generally expand, the granum lamellae become thin, the number of granum lamellae decrease, and the clarity of the capsule and the plasma membranes decreases. With a decrease in temperature and prolonging of stress, high-intensity chilling stress can lead to a series of symptoms, including the darkening of chloroplast stroma, the de-layering of grana, the disintegration of thylakoid membranes and the chloroplast envelope, the accumulation of vesicles, and the disintegration of chloroplasts [92,93,94].

Progress has been made in the study of chloroplast responses to abiotic stresses. However, it is still unclear whether these responses are inevitable consequences of chilling stress induction or are actively performed to induce cold tolerance regulation to avoid damage. Therefore, the causal relationship between chloroplast resistance and the regulation of stress remains to be further studied.

## Figures and Tables

**Figure 1 ijms-20-05046-f001:**
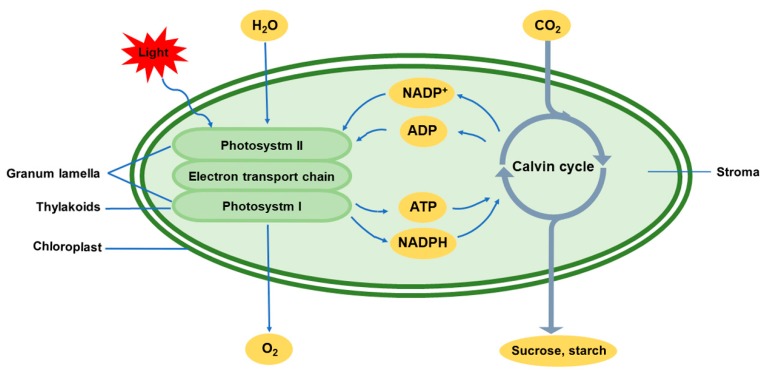
Photosynthesis in chloroplasts under normal conditions. The light reactions of photosynthesis are carried out in the thylakoids of chloroplast. Through absorption and transmission of light energy by photosynthetic pigments, water is broken down into hydrogen and O_2_ by the action of a portion of the light energy. Oxygen atoms are combined to form O_2_, which is released. Electrons released by the oxidation of water molecules are transferred to NADP^+^ by the electron transport chain, and hydrogen is combined with NADP^+^, reducing it to NADPH. Another result of electron transfer is that protons in the stroma are pumped into the thylakoid cavity, forming a trans-membrane proton gradient that drives the phosphorylation of ADP to ATP. The dark reactions of photosynthesis are carried out in the chloroplast stroma. NADPH and ATP produced by the light reactions are used for carbon assimilation in the Calvin cycle, and CO_2_ is reduced to sugars. ADP, adenosine diphosphate; ATP, adenosine triphosphate; CO_2_, carbon dioxide; NADPH, nicotinamide adenine dinucleotide phosphate; NADP^+^, oxidized form of NADPH; O_2_, oxygen.

**Figure 2 ijms-20-05046-f002:**
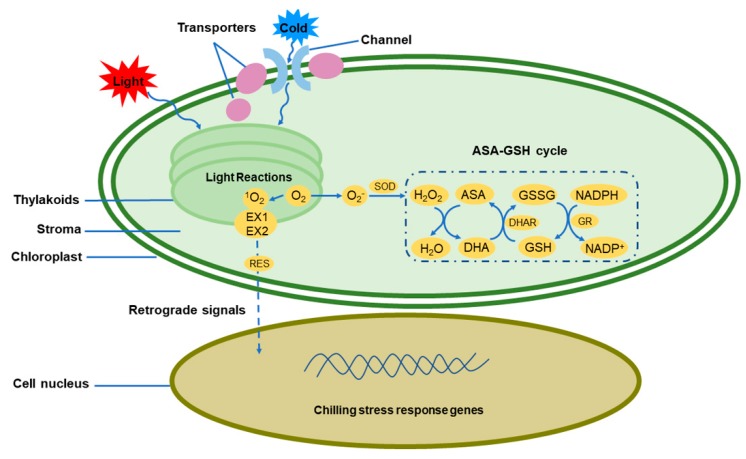
Chloroplast response under chilling stress. Chloroplasts can sense cold stress signals via membrane proteins such as channel proteins or transporter proteins. Under cold stress, excess ROS are produced during the process of photosynthetic electron transport, in which O_2_^−^ is removed by the antioxidant system of SOD and the ASA-GSH cycle. ^1^O_2_ binds to the calcium-sensitive receptor proteins EX1 and EX2 on the thylakoid membrane and acts as a retrograde signal to the nucleus to induce the expression of cold tolerance genes. ASA, ascorbic acid; DHA, dehydroascorbate; DHAR, dehydroascorbate reductase; EX1, EXECUTER1; EX2, EXECUTER2; GR, glutathione reductase; GSH, glutathione; GSSG, oxidized glutathione; H_2_O_2_, hydrogen peroxide; NADPH, nicotinamide adenine dinucleotide phosphate; NADP^+^, oxidized form of NADPH; RES, Reactive electrophile species; ^1^O_2_, singlet oxygen; O_2_, oxygen; O_2_^−^, superoxide anion; SOD, superoxide dismutase.

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
