# Peer review of "Chloroplasts— Beyond Energy Capture and Carbon Fixation: Tuning of Photosynthesis in Response to Chilling Stress"

_ijms, 2019, doi:10.3390/ijms20205046_

Round 1

Reviewer 1 Report

Manuscript titled: “Chloroplasts – Beyond Energy Capture and Carbon Fixation: Tuning of Photosynthesis in Response to Chilling Stress” by Ping Gan and coworkers, submitted to International Journal of Molecular Sciences gives an overview on the response of chloroplasts to chilling stress, emphasizing  the regulatory mechanism of plants in response to low temperature stress via signaling pathways which modulate photosynthesis.  This is an important issue in broad area of the relation of plant and environment. As opposed to most of other papers  where chloroplasts are presented as performing solar energy capture and carbon fixation this manuscript concentrates  on a different role of chloroplasts. Therefore I consider the topic of this manuscript to be interesting.

 Abstract is written in a clear and comprehensive way and reflects the content of the manuscript.

Composition of subsequent chapter  of the manuscript is logically constructed:

Ad. 1 The Introduction starts with description of the chilling stress in general, then of the chloroplast role,  followed by plant hormone role.  Finally, ROS and redox signals produced by chloroplasts  are described as the critical components of plant stress response.

Ad.2  Perception of Chloroplasts to Chilling Stress Signals gives short picture of physiological and biochemical responses of chloroplasts to chilling stress which may be sensed by lipid membrane systems and photoreceptors.

Ad. 3 The Regulatory Response to Chilling Stress Signals in chloroplasts shows, in a nice way, how chloroplasts  are involved in the integration of different intracellular signals and metabolic pathways.  In the part concerning the role of lipids in the response of chloroplasts to chilling stress I would recommend to cite a paper dealing with lipid composition of thylakoid membrane to low temperature: SkupieĹ„ et al.2017, Plant Physiol Biochem 111, 107-118. The following part  of this chapter shows  the importance of chloroplast proteins in adaptation of chloroplasts to chilling stress as well as the role of some photosynthetic enzymes, involved in the Calvin cycle, to compensate enzymes which have reduced activity due to chilling stress. An interesting part of this long chapter is devoted to amplified action of antioxidant systems in chloroplasts  which protect structure and photosynthetic function during low temperature stress. Recent results on the various mechanisms of retrograde signals from the chloroplast to the nucleus are well presented. Last section of the Chapter 3 present the role of plants hormones, especially  as mediators in low temperature tolerance.

Ad. 4 Conclusion and perspectives. This chapter summarizes the subject in a elegant way. However I would not say that „Environmental stresses are great challenges to plant growth and development”. Environmental stresses induce special response which probably would not happened in normal conditions. It is not justified to name them “challenges”.

Although most of the issues described in the manuscript can be found elsewhere the fact that they are collected together results in interesting  review. The manuscript is clearly and nicely written; I read it with pleasure.

Effect of low temperature on chloroplasts  is well known and  well documented subject. Therefore it is not possible, and also not necessary to refer to every paper dealing with this issue. Also I would advise to include data from more recent original papers of last years. Only ¼ about one quarter of cited papers are from the last 5 years.

Concluding: I highly evaluate the results presented in the manuscript “Chloroplasts – Beyond Energy Capture and Carbon Fixation: Tuning of Photosynthesis in Response to Chilling Stress” by Ping Gan and coworkers.  

The subject of the manuscript is interesting and important for cell biologist, physiologists and molecular biologists dealing with different aspects of plant response to environmental stresses.

I recommend the manuscript to be published in the International Journal of Molecular Sciences after the minor correction mentioned above is introduced. 

Author Response

Dear Editor,

Hope you are doing well!

We greatly appreciate for your kindly and helpful suggestions to our manuscript entitled ‘Chloroplasts – Beyond Energy Capture and Carbon Fixation: Tuning of Photosynthesis in Response to Chilling Stress’. We carefully revised the manuscript according to the comments, and now resubmit revised manuscript for your consideration. In the revised version, all detailed revisions were described in ‘Point-by-point response to reviewers’ shown below for the convenience of you viewing.

We think the current version of this manuscript would have been significantly improved after revision.

Thanks again and any further suggestions to our manuscript from you will be highly appreciated and we are very grateful for your kind consideration for publication.

Best wishes

Yours sincerely,

Jijing Luo, Professor (Ph.D.)

College of Life Science and Biotechnology, Guangxi University

State Key Laboratory for Conservation and Utilization of Subtropical Agro-bioresources, Guangxi University

100 Daxue Rd. (East), Nanning, 530004, China

Cell phone: +86-18077792389

Point-by-point response to reviewers

We thank the Editor and Reviewers for insightful comments, which help us to substantially revise the manuscript.

In the current version of manuscript, we made some revisions according to the comments, including English language editing, supplemented References, and adjusted the description of the related content for more readability.

Reviewers’ suggestions for several descriptions were adopted for more readability.

Notes: Based on the guideline of editorial office, any revisions should be clearly highlighted. We have used the "Track Changes" function in Microsoft Word to modify the manuscript, and all the detailed revisions to the manuscript were described in this section.

Comments and Suggestions:

In the part concerning the role of lipids in the response of chloroplasts to chilling stress I would recommend to cite a paper dealing with lipid composition of thylakoid membrane to low temperature: Skupień et al.2017, Plant Physiol Biochem 111, 107-118.

Reply: Thank you for your careful reading and great suggestion! The paper you recommended provides a strong basis for us to elucidate the regulation of lipid membrane status of chloroplasts at low temperature. So, we added a paragraph in section 3.1. (p4, line28-35, revised manuscript)

Conclusion and perspectives. This chapter summarizes the subject in a elegant way. However I would not say that “Environmental stresses are great challenges to plant growth and development”. Environmental stresses induce special response which probably would not happened in normal conditions. It is not justified to name them “challenges”.

Reply: Thank you for your careful reading and great suggestion! According to your suggestions, we have replaced the word “challenges” and changed it to “influence factors”, which would be more appropriate. (p9, line27, revised manuscript)

I would advise to include data from more recent original papers of last years. Only ¼ about one quarter of cited papers are from the last 5 years.

Reply: Thank you for your careful reading and great suggestion! We have supplemented the references to improve the proportion of references cited in the past five years, but there are few recent relevant studies, and references that fit the theme of this paper have been basically cited. Supplementary references are as follows:ref.24, ref.25 and ref.34.

Reviewer 2 Report

In this review paper Gan et al survey the current state of knowledge regarding the responses of chloroplast membranes to low temperature stress and the signaling pathways that modulate the response to this stress by way of changes to the existing proteome and gene expression.  It is a good review and I recommend it for publication if some minor changes are made.  Some changes to the language in the first three pages would help readers.  I found no problems with the remainder of the paper.

Minor changes:

1, line 38. The word “stress” is overused. I suggest “Low-temperature conditions, including chilling and freezing, are environmental stresses that can…”

1, lines 39-42. Sentence beginning with “Although freezing stress…”. There is no need to explain that freezing stress will not be covered.  Instead, just say there will be a focus on chilling stress.  I suggest replacing the sentence with something like, “In this review we focus on the response mechanisms of chloroplasts to chilling stress because many crop plants experience this during the growing season.”

2, line 2. “…temperature is an important…”

p.2, line 7. “Approximately seven million hectares in South and…”

p2. Lines 9-10.  The language at the end of this sentence is awkward, I recommend changing it to… “…modern rice varieties cannot be planted due to the cool climate inherent to these regions [4].”

p3. Line 33.  I recommend, “Here, we briefly review how chloroplasts perceive chilling stress…”

p3. Line 35.  I recommend removing “their functions in” so it says, “…in addition to energy capture and carbon fixation.”

P3. Lines 37-38.  I recommend removing the last part of the sentence so it reads,  “…which acts as a fine-tuning regulatory mechanism.”

p.3 line 46. “…protein synthesis damages the integrity of photosynthetic membranes thereby impairing the light…”

Author Response

Dear Editor,

Hope you are doing well!

We greatly appreciate for your kindly and helpful suggestions to our manuscript entitled ‘Chloroplasts – Beyond Energy Capture and Carbon Fixation: Tuning of Photosynthesis in Response to Chilling Stress’. We carefully revised the manuscript according to the comments, and now resubmit revised manuscript for your consideration. In the revised version, all detailed revisions were described in ‘Point-by-point response to reviewers’ shown below for the convenience of you viewing.

We think the current version of this manuscript would have been significantly improved after revision.

Thanks again and any further suggestions to our manuscript from you will be highly appreciated and we are very grateful for your kind consideration for publication.

Best wishes

Yours sincerely,

Jijing Luo, Professor (Ph.D.)

College of Life Science and Biotechnology, Guangxi University

State Key Laboratory for Conservation and Utilization of Subtropical Agro-bioresources, Guangxi University

100 Daxue Rd. (East), Nanning, 530004, China

Cell phone: +86-18077792389

Point-by-point response to reviewers

We thank the Editor and Reviewers for insightful comments, which help us to substantially revise the manuscript.

In the current version of manuscript, we made some revisions according to the comments, including English language editing, supplemented References, and adjusted the description of the related content for more readability.

Reviewers’ suggestions for several descriptions were adopted for more readability.

Notes: Based on the guideline of editorial office, any revisions should be clearly highlighted. We have used the "Track Changes" function in Microsoft Word to modify the manuscript, and all the detailed revisions to the manuscript were described in this section.

Comments and Suggestions:

p1, line 38. The word “stress” is overused. I suggest “Low-temperature conditions, including chilling and freezing, are environmental stresses that can…”

Reply: It was significantly improved after your revision for the sentence. Thank you! (p1, line38-39, revised manuscript)

p1, lines 39-42. Sentence beginning with “Although freezing stress…”. There is no need to explain that freezing stress will not be covered. Instead, just say there will be a focus on chilling stress. I suggest replacing the sentence with something like, “In this review we focus on the response mechanisms of chloroplasts to chilling stress because many crop plants experience this during the growing season.”

Reply: Thank you for your careful reading and excellent revision. (p1, line39-43, revised manuscript)

p2, line 2. “…temperature is an important…”

Reply: Thank you for your careful reading and excellent revision. (p2, line3, revised manuscript)

p2, line 7. “Approximately seven million hectares in South and…”

Reply: Thank you for your careful reading and excellent revision. (p2, line9, revised manuscript)

p2, lines 9-10. The language at the end of this sentence is awkward, I recommend changing it to… “…modern rice varieties cannot be planted due to the cool climate inherent to these regions [4].”

Reply: It was significantly improved after your revision for the sentence. Thank you! (p2, line10-12, revised manuscript)

p3, line 33. I recommend, “Here, we briefly review how chloroplasts perceive chilling stress…”

Reply: Thank you for your careful reading and excellent revision. (p3, line35, revised manuscript)

p3, line 35. I recommend removing “their functions in” so it says, “…in addition to energy capture and carbon fixation.”

Reply: Thank you for your careful reading and excellent revision. (p3, line37, revised manuscript)

p3, lines 37-38. I recommend removing the last part of the sentence so it reads, “…which acts as a fine-tuning regulatory mechanism.”

Reply: It was significantly improved after your revision for the sentence. Thank you! (p3, line39-41, revised manuscript)

p3, line 46. “…protein synthesis damages the integrity of photosynthetic membranes thereby impairing the light…”

Reply: Thank you for your careful reading and excellent revision. (p3, line49, revised manuscript)